# Microbial Species–Area Relationships in Antarctic Cryoconite Holes Depend on Productivity

**DOI:** 10.3390/microorganisms8111747

**Published:** 2020-11-07

**Authors:** Pacifica Sommers, Dorota L. Porazinska, John L. Darcy, Eli M. S. Gendron, Lara Vimercati, Adam J. Solon, Steven K. Schmidt

**Affiliations:** 1Department of Ecology and Evolutionary Biology, University of Colorado Boulder, Boulder, CO 80309, USA; lara.vimercati@colorado.edu (L.V.); adam.solon@colorado.edu (A.J.S.); steve.schmidt@colorado.edu (S.K.S.); 2Department of Entomology and Nematology, University of Florida, Gainesville, FL 32611, USA; dorotalp@ufl.edu (D.L.P.); eli.gendron@colorado.edu (E.M.S.G.); 3Division of Biomedical Informatics and Personalized Medicine, University of Colorado School of Medicine, Aurora, CO 80045, USA; darcyj@colorado.edu

**Keywords:** Antarctica, bacteria, biogeography, cryoconite, eukaryotes, glacier, ISAR, species–area relationship

## Abstract

The island species–area relationship (ISAR) is a positive association between the number of species and the area of an isolated, island-like habitat. ISARs are ubiquitous across domains of life, yet the processes generating ISARs remain poorly understood, particularly for microbes. Larger and more productive islands are hypothesized to have more species because they support larger populations of each species and thus reduce the probability of stochastic extinctions in small population sizes. Here, we disentangled the effects of “island” size and productivity on the ISAR of Antarctic cryoconite holes. We compared the species richness of bacteria and microbial eukaryotes on two glaciers that differ in their productivity across varying hole sizes. We found that cryoconite holes on the more productive Canada Glacier gained more species with increasing hole area than holes on the less productive Taylor Glacier. Within each glacier, neither productivity nor community evenness explained additional variation in the ISAR. Our results are, therefore, consistent with productivity shaping microbial ISARs at broad scales. More comparisons of microbial ISARs across environments with limited confounding factors, such as cryoconite holes, and experimental manipulations within these systems will further contribute to our understanding of the processes shaping microbial biogeography.

## 1. Introduction

One of the most ubiquitous patterns in ecology is the island species–area relationship (ISAR). ISAR patterns were originally described in the context of plants and animals [1,2,3]. They have since been shown to apply to archaea, bacteria, fungi, and other microscopic eukaryotes in host-associated and free-living environments ranging from the temperate to the extreme [4,5,6,7,8,9,10,11,12,13]. Interest from ecologists in ISARs has been driven in part by its implied applications in forecasting the effects of habitat fragmentation and climate change [14,15,16]. Use of island-like habitats of microbes has generated interest both for their potential as model systems and for understanding how the characteristics of microbes and their communities may affect their biogeography [17,18].

Productivity of the environment is thought to significantly shape ISARs, although empirical evidence for how productivity and area interact with species richness has varied with the region and sampling scale of studies [19,20,21,22]. This interaction has not yet been studied for microbes under controlled, comparable conditions. Larger island-like habitats, and those with greater productivity, are hypothesized to be richer in species because they support larger populations, meaning each species has a larger population and is less vulnerable to stochastic extinction (“drift”) [23,24,25]. By contrast, if the same number of species were present on a much smaller island that supported fewer individuals overall, the populations of each species would be much smaller and, therefore, more vulnerable to stochastic extinctions, reducing the number of species expected to persist there [1,23]. Besides the total number of individuals and number of species, the relative evenness in the individuals per species can also affect the shape and fit of the ISAR [21]. Uneven populations could have implications both for the viability of the rare populations and the probability of detecting them [26,27]. The contributions of productivity and community evenness to microbial ISARs have yet to be explored across broad taxonomic groups within similar environments.

We used Antarctic cryoconite holes as a model system to ask how productivity and evenness shape ISARs for bacteria and for microbial eukaryotes. Cryoconite holes are island-like habitats whose similarity in physical conditions but variation in both size and in productivity make them amenable to serve as natural mesocosms for understanding drivers of microbial diversity [28,29] (Figure 1). Cryoconite holes form when low-albedo sediment is blown onto the surface of a glacier, absorbs solar radiation, and melts into the ice [30,31,32,33]. The sediment contains communities of bacteria and microbial eukaryotes [33,34,35,36], whose community diversities are positively correlated [37]. During the brief Antarctic summer, these microbes take advantage of liquid water inside the cryoconite hole to grow [38,39,40]. An entire food web of heterotrophic bacteria, ciliates, tardigrades, and rotifers is supported by primary production from cyanobacteria and microalgae [33]. Photosynthetic activity can be so great that the produced dissolved oxygen leads to extremely high pH conditions when atmospheric equilibration is prevented by the formation of an ice lid [38,41,42] (Figure 1). Ice-lidded cryoconite holes are unique to Antarctica, where they can remain isolated from one another and from the atmosphere for a decade or more [31].

Antarctic cryoconite holes make compelling systems for the study of microbial ISARs because they are island-like systems [43] and the holes have very similar physical conditions. Island-like systems are discontinuous patches whose diversity varies with their size and whose insularity levels vary with distance from one another, both of which are true of cryoconite holes [5]. Cryoconite holes in the McMurdo Dry Valley region of Antarctica typically range from 5 to 145 cm in diameter [31]. The area of a cryoconite hole directly correlates with the mass of the sediment inside, and the addition of more sediment increases the hole’s diameter to maintain a relatively consistent sediment depth [44]. The degree of connectivity among cryoconite holes, which depends on the contrast between the “matrix” and “islands” on glaciers, can vary among regions: in alpine and Arctic glaciers, the entire surface of the glacier can become a slushy weathering crust during the summer that can facilitate the movement of microbes [45]. In contrast, Antarctic cryoconite holes can remain isolated without hydrological connectivity for decades [31], which would allow many generations of microbes to elapse in isolated conditions. Furthermore, previous research on Antarctic cryoconite holes, albeit over a limited range of hole sizes, indicated that bacteria in cryoconite holes showed a significant ISAR [5].

Here, we used amplicon sequencing of bacteria and microbial eukaryotes within cryoconite holes over a larger range of sizes than previously studied on two glaciers that differ in productivity [35]. We hypothesized (1) that more species would accumulate with hole area (island size) on the more productive Canada Glacier than on the less productive Taylor Glacier; (2) that additional variation in the ISAR within each glacier would be related to measures of productivity such as DNA concentration because more productive holes would essentially function as biologically “larger” islands than their physical size would suggest [46]; and (3) that additional variation in the ISAR within each glacier would be related to community evenness because holes in which a small number of species dominate the community would have fewer rare species detected and, thus, lower richness detected.

## 2. Materials and Methods

We collected samples from frozen cryoconite holes on Canada (77.61913° S, 162.937283° E) and Taylor (77.73033° S, 162.05768° E) glaciers in Taylor Valley, Antarctica, in December 2017. Logistical constraints prevented the collection of as many samples from Taylor Glacier as from Canada Glacier. Cryoconite holes on Canada Glacier (23 total) ranged from 7 to 95 cm in diameter, and those on Taylor Glacier (11 total) from 15 to 95 cm in diameter. All sampled holes on each glacier were within 500 m of one another. Spatial autocorrelation of community composition has not been detected at that scale for these glaciers [5]. A 10-cm diameter SIPRE corer was used to extract a “puck” of frozen sediment from each cryoconite hole (Figure 1), which was transferred to a sterile polyethylene bag using nitrile gloves cleaned with 70% isopropyl alcohol. We brought the samples to a field laboratory within six hours of collection and stored them at −20 °C for one to three weeks before melting, homogenizing, and subsetting them for analysis. To prevent cross-contamination among cryoconite holes, we drilled into bare glacial ice (containing no visible sediment) prior to sampling each cryoconite hole. In addition, the outer layer of ice and sediment was melted off each sampled sediment puck by using filtered, deionized water in the laboratory before each sample was melted and homogenized.

Each cleaned sediment puck was placed in a beaker sterilized with 70% ethanol and UV light and stored at 4 °C for 12–24 h to melt. When sediments were incompletely melted after that time, they were placed at room temperature for up to 4 h for final melting. Sediments were mixed and allowed to settle before excess meltwater was poured off. We then subset ~0.3 g sediments for DNA extraction and ~5 g for determining water content and organic matter content.

Because the entire sediment mass of each cryoconite hole could not be collected, especially in the larger holes where the diameter of the hole was larger than the diameter of the corer, we quantified the total dry mass of collected and homogenized sediment. We calculated the water and organic matter content of each sample by weighing the 5-g subsets before and after drying them at 60 °C for 24 h and again following combustion at 450 °C for 4 h in a muffle furnace (ash-free dry mass, a measure of organic matter content). The dry equivalent mass of collected and homogenized sediment was then used as a possible covariate related to variation in the richness–area relationship, as described below.

We extracted DNA with the DNeasy Power Soil DNA Extraction Kit (Qiagen, Hilden, Germany) and measured the concentration of DNA in each using a Qubit fluorometer (Thermo Fisher Scientific, Waltham, MA, USA), all according to the manufacturers’ directions. We stored extracted DNA at or below −20 °C for up to five months before amplifying and sequencing. Each sample was amplified with the Earth Microbiome Project primers for 16S and 18S small subunit rRNA genetic markers, 515f-806r [47,48] and 1391-EukBr [49], to detect bacteria and eukaryotes, respectively. Amplified DNA was pooled and normalized to equimolar concentrations using SequalPrep Normalization Plate Kit (Thermo Fisher Scientific, Waltham, MA, USA). Libraries were sequenced on an Ilumina MiSeq at 2 × 250 bp chemistry for the 16S library and 2 × 150 bp chemistry for the 18S library by the Biofrontiers Sequencing Facility at the University of Colorado Boulder.

To process the sequence data, we first demultiplexed the data using idemp and trimmed the primers with Cutadapt [50]. After visual inspection of quality plots, we trimmed 16S forward reads to 200 bp and reverse reads to 160 bp, with no more than 2 expected errors for each. Furthermore, 18S forward reads were trimmed to 125 and reverse reads to 120 with no more than 2 expected errors. We used ‘dada2′ to denoise the data and infer 100% exact sequence variants (ESVs) and then to merge the paired-end reads and remove chimeras [51,52]. Taxonomy was assigned against the Silva 132 NR99 reference database using the parallel_assign_taxonomy_blast.py command from the QIIME workflow [53,54]. The taxonomic assignments of the most abundant ESVs from each sample were verified by using BLAST (NCBI online tool). We used the ‘mctoolsr’ package to filter out chloroplasts, mitochondria, and eukaryotic reads from the 16S data and bacterial reads from the 18S data [55]. After visual inspection of ESVs present in the samples and negative controls, sequences prevalent in the negative controls were removed. In the following analyses, we did not rarefy the sequence reads for several reasons. First, an increasing body of literature documents that rarefaction is inappropriate as it leads to severe underestimation of the presence and abundance of rare taxa, which, in our study, we particularly wanted to observe in order to obtain accurate estimates of richness [27,56]. Because unrarefied reads may contribute to the variation in ESV richness due to differences in PCR [27], we therefore tested whether the number of reads recovered from sequencing per g sediment contributed to richness–area models as described below.

The most commonly used model for the ISARs is the power law,
(1)$$S=cA^{z},$$
where *S* is the number of species, *A* is the area of the island, and *c* and *z* are parameters estimated empirically that determine the shape of the ISAR [57,58]. As such, hypotheses regarding the shape of ISARs typically focus on *z* and, to a lesser extent, on *c.* One reason the power law model is so popular in island biogeography is because of its convenient form as a linear regression on the logarithmic scale,
(2)$$log_{10}S=log_{10}c+zlog_{10}A.$$

The logarithm of the data can be used to estimate a linear intercept, which corresponds to log_10_*c*, and slope, which corresponds directly to the value of *z*. However, referring to these parameters as the slope and intercept of the ISAR directly can be misleading because, on the linear scale, both parameters actually affect the rate at which islands accumulate species.

In keeping with much of the literature, we therefore used Equation (2) to estimate the parameters *c* and *z* for our data. To test our first hypothesis, that cryoconite holes on the more productive Canada Glacier would accumulate more species of both bacteria and eukaryotes with increasing sizes than holes on the less productive Taylor Glacier, we used an analysis of covariance (ANCOVA) in R v 3.6.2 [59], with the number of unique ESVs in a sample as a proxy for species, *S*. We used the R package ‘car’ to calculate the Type III errors [60], in which the variation assigned to each factor is not conditional on the order of terms in the model. We then estimated the values and standard errors of model parameters from fitting linear models separately to the data from each domain of life on each glacier. All plots were made with R packages ‘ggplot2’ and ‘ggpubr’ [61,62].

To test our second hypothesis, that productivity would relate to additional variation in the ISAR within each glacier, we added DNA concentration as a covariate to the linear regression between log_10_*S* against log_10_*A* after accounting for glacier. Productivity by design varied between the two glaciers to test our first hypothesis, and so we used Type I errors to evaluate the factors in these models sequentially: that is, to ask what the additional explanatory power of DNA concentration on log_10_*S* was after accounting for the effect of hole area and glacier.

To test our third hypothesis, that evenness in the relative abundances would explain additional variation in the ISAR within each glacier, we calculated the inverse Simpson index for each sample using the R package ‘hillR’ as a measure of community evenness [63]. Similar to our approach to testing the second hypothesis, we used sequential partitioning of variance (Type I errors) to test whether adding the evenness metric to the linear regression of log_10_
*S* against log_10_*A* was significantly related to additional variation after accounting for glacier. Additionally, we performed an ANCOVA on evenness with log_10_*A* to determine whether it co-varied with area and, similar to our approach for characterizing the ISAR, whether evenness differed between glaciers when accounting for area.

Finally, we also tested whether metrics related to the total mass of collected and homogenized sediment per sample, or the number of sequences obtained per sample, explained the ISAR—in addition to hole area and glacier. Area sampled is related to the richness of species detected in many systems, similar to the ISAR, although the processes resulting in the pattern are distinct from isolated island-like systems. Similarly, the reads obtained per g sediment could be interpreted as uneven sampling effort, biasing the results, especially in cases where the variation in reads primarily reflects stochastic variation in the PCR amplification step rather than underlying biological differences. Details of those analyses and results are in Appendix A.

## 3. Results

Amplicon sequencing resulted in 748,604 quality-filtered, paired-end reads for the 16S SSU rRNA genes (Bacteria) assigned to 2133 ESVs across the 34 samples and 764,388 18S SSU rRNA genes (Eukarya) assigned to 458 ESVs. The 16S primers also detected archaea but fewer than 0.02% of the reads were assigned to archaea, so we refer to 16S results as profiling bacterial communities for simplicity. Sequence data and associated metadata are available in the NCBI’s Sequence Read Archive under PRNJNA668398. The bacterial communities were dominated by Cyanobacteria, particularly on Canada Glacier, with some Alphaproteobacteria, Bacteroidetes, and Actinobacteria also in the ten most relatively abundant ESVs (Figure 2). Eukaryotic communities were dominated by microalgae (Chlorophyta), bdelloid rotifers, and tardigrades, but also by diatoms (Bacillariophyta) and Platyhelminthes (Figure 2).

Our first hypothesis was supported, that more species accumulated with hole area on the more productive Canada Glacier than on the less productive Taylor Glacier (Figure 3). The intercept of the ISAR on the log_10_ scale differed significantly between glaciers for both bacteria (ANCOVA: *F*_DF=1_ = 12.3, *p* = 0.001) and eukaryotes (ANCOVA: *F*_df=1_ = 120.1, *p* < 0.001), with no significant interaction between area and glacier on the log_10_ scale (bacteria: *F*_DF=1,30_ = 1.6, ns; eukaryotes: *F*_DF=1,30_ = 2.1, ns). Deviations of the model’s residuals from normality were not observed (Shapiro–Wilk test for normality in Canada bacteria: *W* = 0.96, ns; Taylor bacteria: *W* = 0.86, ns; Canada eukaryotes: *W* = 0.96, ns; Taylor eukaryotes: *W* = 0.93, ns), and variances were even as well (Levene test for unequal variances in bacteria: *F*_(DF=1,32)_ = 1.40, ns; eukaryotic *F*_(DF=1,32)_ = 0.56, ns). The significant difference observed in intercepts between Canada and Taylor glaciers’ ISARs on the log_10_ scale indicates that the shape of the ISARs is significantly different on a linear scale (Figure 3b). On a linear scale, species richness is scaled by a larger value of *c* and accumulates more rapidly in smaller holes on the high-biomass Canada Glacier than it does on the lower-biomass Taylor Glacier for both bacteria and microbial eukaryotes.

An ANOVA revealed a significant interaction between the area and domain, indicating that the slope of the ISAR did differ by domain (*F*_DF=1_ = 7.88, *p* = 0.006). Estimates of the slope and intercept of linear models on the log_10_ scale for each domain on each glacier are given in Table 1 and an approximation of the derivative of each slope on a linear scale is illustrated in Figure A1 (Appendix B).

Neither productivity nor community evenness were consistently significantly related to deviations from the ISAR (Figure 4 and Figure 5). After accounting for area and glacier, DNA concentration did not explain significantly more variation for bacteria (*F*_DF=1_ = 0.82, ns) or for eukaryotes (*F*_DF=1_ = 0.65, ns), nor was it significantly colinear with the log_10_ area of cryoconite holes (Appendix B). Evenness, as measured by the inverse Simpson index, also did not contribute significantly to the richness–area relationship model for bacteria (*F*_DF=_ = 0.024, ns) or for eukaryotes (*F*_DF=_ = 0.13, ns), perhaps because evenness also increased with area for both bacteria (*F*_DF=1,31_ = 43.9, *p* < 0.001) and eukaryotes (*F*_DF=1,31_ = 43.9, *p* < 0.001) (Figure 5). Interestingly, evenness was higher on Canada Glacier than Taylor Glacier for eukaryotes (*F*_DF=1,31_ = 9.01, *p* = 0.005), but the reverse was true for bacteria (*F*_DF=1,31_ = 13.1, *p* = 0.001), without any significant interaction between area and glacier (bacteria: *F*_(DF=1,30)_ = 0.27, ns; eukaryotes: *F*_(DF=1,30)_ = 1.51, ns), or violations of normality in residuals (Shapiro–Wilk test for normality in Canada bacteria: *W* = 0.94, ns; Taylor bacteria: *W* = 0.90, ns; Canada eukaryotes: *W* = 0.97, ns; Taylor eukaryotes: *W* = 0.85, ns) or variance structures (Levene test for unequal variances in bacteria: *F*_(DF=1,32)_ = 0.52, ns; eukaryotic *F*_(DF=1,32)_ = 3.37, ns).

## 4. Discussion

Despite ISARs having been characterized in microbes for more than fifteen years now [10,11], the processes underlying such ubiquitous patterns and the causes of deviations from them have not been resolved. We showed that microorganisms inhabiting Antarctic cryoconite holes exhibited ISARs that differed with productivity for two major domains of life, bacteria and microbial eukaryotes. Cryoconite holes on the more productive Canada Glacier accumulated more species with hole area than on the less productive Taylor Glacier and supported our first hypothesis, that both productivity and area contribute to microbial diversity. Many more species of bacteria than eukaryotes were found in every size of cryoconite hole and on both glaciers, and the shape of the ISAR differed between the two domains. Despite these differences, the greater number of species on Canada Glacier than Taylor Glacier was consistent across both domains, emphasizing the generality of our findings. Within each glacier, additional variation was not attributable to differences in productivity or community evenness, contrary to our second and third hypotheses. Our results, overall, are consistent with both area and productivity increasing species richness on a broad scale, but do not explain smaller deviations from ISARs. Smaller deviations within our experiment may not have been explained because of the strong fit of the species–area power law to our data, leaving little residual behind to be explained by other factors.

The dominant members of both bacterial and eukaryotic communities overlapped substantially between glaciers, although the biomass concentrations of the two glaciers differed significantly, in line with previous results [34,35,37]. The abundance of microinvertebrates and overall biomass varied between Canada and Taylor Glaciers, reflecting the makeup of the streams and soils surrounding the glaciers [34,35,64]. Sequences matching tardigrades and bdelloid rotifers were relatively abundant on both glaciers, although the actual abundance of all organisms was likely much higher on Canada Glacier, as evidenced by the differences in DNA concentration extracted. Tardigrades and rotifers are commonly found in cryoconite holes worldwide [65,66]. Consistent with previous studies of cryoconite holes in Taylor Valley and other locations, phototrophs, including cyanobacteria, microalgae, and diatoms, dominated the most relatively abundant sequence variants of both bacteria and eukaryotes in these net photosynthetic systems [35,37,38]. Microbial mats from the nearby streams, consisting of cyanobacteria and algae, have been suggested as a likely source for cryoconite on these glaciers, and their dominance may be indicative of those potential sources [67,68]. Future work should further investigate the role of the composition of potential sources of cryoconite on the communities that develop within cryoconite holes.

Despite the many sources and environmental characteristics of microbial island-like systems that have been studied around the world, as well as the methodology used to sample them, the power law exponent, *z*, of microbial ISARs around the world falls within a similar range [4,12,13]. The exponents we found in Antarctic cryoconite holes were no exception. Most studies of microbial ISARs, using a variety of taxonomic groups and molecular methods, have found a *z* (which corresponds to slope in the linearized equation) of ~0.2, with a range of ~0.02 to ~0.5, more than an order of magnitude [4]. Our estimated exponents for bacteria and eukaryotes across two glaciers ranged from 0.16 to 0.31, which also overlaps with that of some non-microbial taxonomic groups in true island archipelagos, which are typically 0.2–0.4 [24,57], although *z* values have been found for some groups of animals as high as 0.7 [57].

The other key parameter, besides *z*, in the power law ISAR which has received considerably less attention is the scaling constant, *c* [24]. The scaling parameter has been argued to reflect a number of different properties; for example, a measure of carrying capacity, or a scale-independent measure of diversity, dependent on ecological characteristics of species, such as dispersal capabilities and population abundance [24,57]. The vast differences in *c* between bacteria and eukaryotes found in this study were therefore expected due to the greater abundance and diversity of bacteria in nature. The significant differences between glaciers were also consistent with larger populations on the more productive Canada Glacier than in cryoconite holes on the less productive Taylor Glacier.

Besides overall productivity, evenness in the relative abundance of species affects population sizes in an island-like habitat, with seemingly potential implications for the persistence and detection of rare species in ISAR studies. Very uneven populations could result in rare species going undetected, depending on the sampling depth, or small populations being vulnerable to stochastic catastrophes and resulting extinction [26,27]. The consistent increase in community evenness with area in our study was similar to other work that found significant power law relationships in bacterial evenness–area relationships, typically with lower values of *z* than the species–area relationship [6,69]. The increasing evenness could primarily be a function of the nature of relative abundance data: as more rare taxa are detected in samples in larger island-like habitats, the proportion of reads represented by the most relatively abundant organisms declines, even if their total abundance remains the same or increases, leading measures of evenness to increase. More quantitative studies of absolute abundance will help to clarify the relationship between community structure and the area of cryoconite holes in microbial island-like systems.

The selection of a sampling scale relevant to a given ecological question remains a critical issue in microbial biogeography and microbial ecology generally [9,70], and the area sampled in any site can have a similar species–area relationship to islands, but for different reasons [71]. However, the relationship between species and area in a contiguous environment is more of a question of spatial heterogeneity, dispersal, and sampling, whereas the relationships in island-like systems are also governed by colonization and extinction dynamics given the impermeable matrix between “islands” [58]. Nested plots of varying contiguous area pick up more individuals and thus more rare taxa in larger plots than smaller plots or include a larger variety of environmental conditions that support different species. Because we were interested in capturing the best estimate possible of the taxa present in the island as a whole, we took the approach of aiming to collect and homogenize as much of the sediment from each hole as possible. The mass of sediment per hole collected and homogenized for DNA extraction therefore varied significantly—although not perfectly—with the area of the hole, with variation introduced due to the angle of the core and the limitations of its dimensions (10-cm diameter). Although the scale of sampling may, therefore, introduce an additional element of sampling effect into the ISAR measured here, the sampled area reflected the microbial diversity available in each hole and was therefore the more relevant measurement to characterize the ISAR.

Overall, our results highlight the importance of productivity on microbial ISARs across major domains of life and the potential utility of cryoconite holes as a model study system for better understanding them. As microbial ecology moves from describing patterns to determining the processes that generate patterns, more mechanistic experimental work in cryoconite holes should build on this research to further clarify the processes that influence ISARs.

## Figures and Tables

**Figure 1 microorganisms-08-01747-f001:**
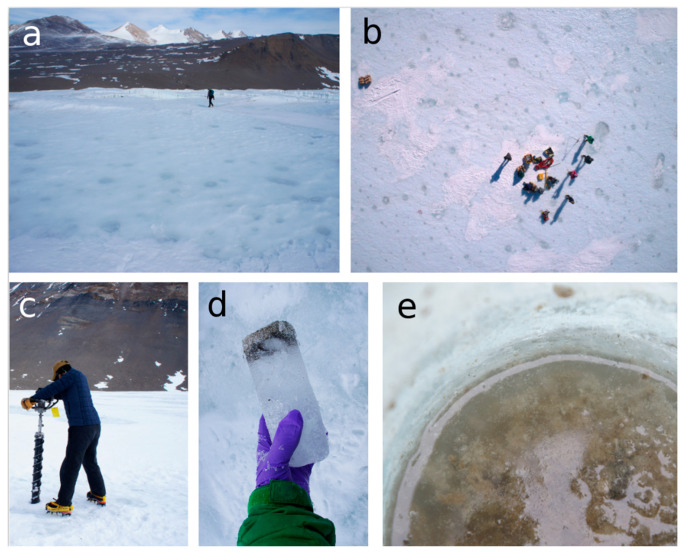
(**a**) Ice-lidded cryoconite holes of variable sizes show up as blue circles in the white surface ice of Canada Glacier. (**b**) Ice-lidded cryoconite holes on the surface of Taylor Glacier as photographed with a research team from above by an uncrewed aerial system (UAS) or drone (photo courtesy of Brendan Hodge). (**c**) Coring through the frozen lid of a cryoconite hole with a SIPRE corer. (**d**) Early in the austral summer, the sediment is still frozen at the base of the holes and can be extracted as a “puck” in a core. (**e**) During the austral summer, the water inside cryoconite holes melts under the ice lid.

**Figure 2 microorganisms-08-01747-f002:**
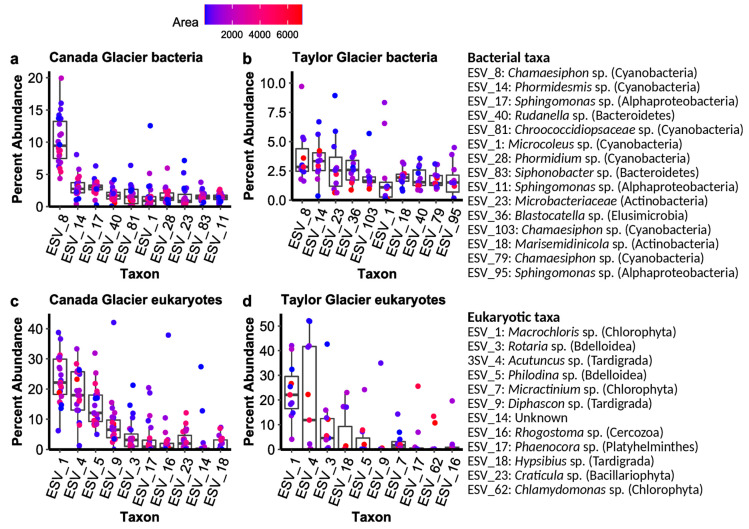
Percent abundance of dominant sequence variants in (**a**) bacteria from Canada Glacier and (**b**) Taylor Glacier cryoconite holes, as well as (**c**) eukaryotes from Canada Glacier and (**d**) Taylor Glacier cryoconite holes. Color for individual hole is scaled to the log_10_ area.

**Figure 3 microorganisms-08-01747-f003:**
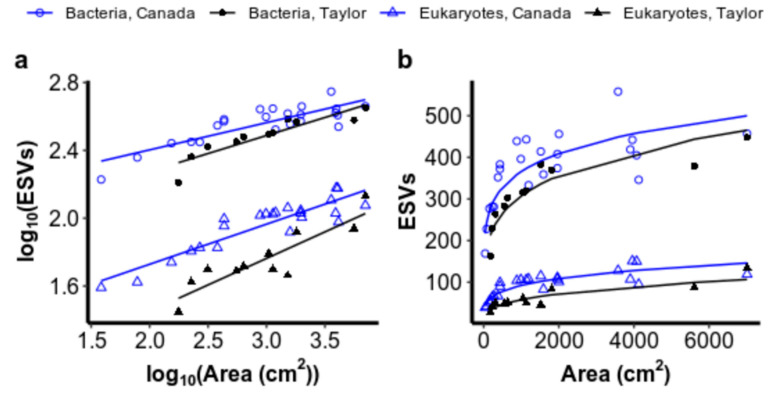
(**a**) Linear relationships between the log_10_*S* as measured by exact sequence variants (ESVs) and the log_10_*A* for bacteria and eukaryotes from cryoconite holes on Canada and Taylor glaciers. (**b**) The same relationships plotted on a linear scale to illustrate the effect of a difference in the value of *c* on species accumulation with area between the glaciers for small cryoconite holes.

**Figure 4 microorganisms-08-01747-f004:**
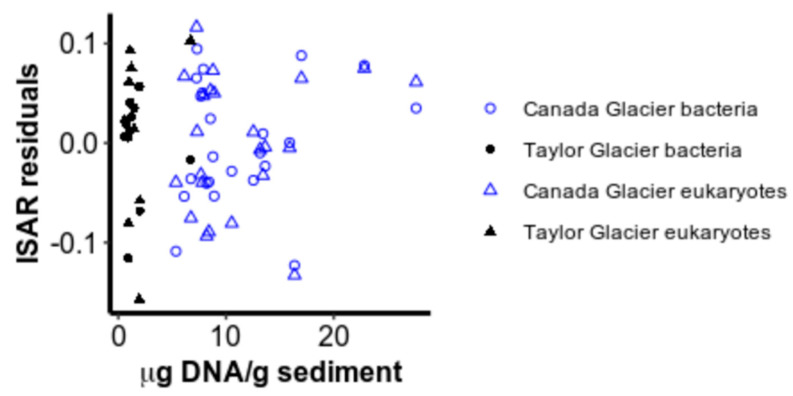
Residuals from the power law island species–area relationship (ISAR) plotted against the DNA concentration extracted from each sample.

**Figure 5 microorganisms-08-01747-f005:**
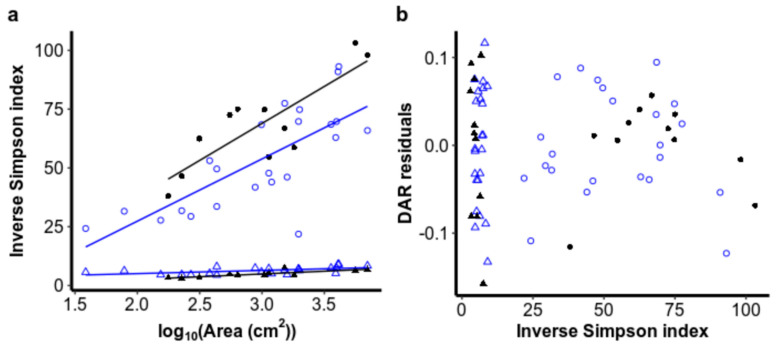
(**a**) Relationships of community evenness, as measured by the inverse Simpson index, regressed against cryoconite hole area. (**b**) Residuals from the richness–area relationship plotted against the community evenness of the same sample.

**Table 1 microorganisms-08-01747-t001:** Parameters estimated from linear models of log_10_ sequence variant richness and log_10_ area of cryoconite hole separately for each domain on each glacier.

Domain	Glacier	Intercept	Std. Err.	Slope	Std. Err.
Bacteria	Canada	2.08	0.066	0.16	0.022
	Taylor	1.85	0.098	0.21	0.032
Eukaryotes	Canada	1.26	0.073	0.24	0.024
	Taylor	0.83	0.16	0.31	0.054

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
