# Peer review of "Microbial Species–Area Relationships in Antarctic Cryoconite Holes Depend on Productivity"

_microorganisms, 2020, doi:10.3390/microorganisms8111747_

Round 1
Reviewer 1 Report
This is an interesting study, using cryoconite holes on Dry Valley glaciers as a 'natural laboratory' to investigate the ecological concept of island species-area. It's a nice idea, and clearly written. Overall, I find the conclusions compelling, but have a couple of comments/suggestions below:
Specific comments/questions to authors
- The main potential drawback to this study is the potential of sampling artefacts (the initial sampling of different masses of cryoconite) to confound the interpretation that species richness correlates with hole size. By the nature of sampling, more material was typically sampled, melted, and homogenised in larger relative to smaller holes prior to taking 0.3 g for DNA extraction and analysis. Could therefore the results be due to this artefact, rather than true hole size? The authors provide an answer to this point in the Appendix, and show that sediment mass does not have as good a linear model fit (R2 = 0.54) to species richness as hole size (R2 = 0.75). To further strengthen this, however, could they also conduct the same statistical analyses but only on those holes which had a diameter greater than that of the corer (as these would presumably have similar masses of sediment in the core but different diameters?) If these also show the same statistical trends (at least on Canada, where hopefully there would be enough holes of sufficient size for this), then this would add another level of robustness to the key conclusions.
- I was also wondering why different numbers of holes were sampled on each of the two glaciers (23 on Canada, 11 on Taylor) - was this time constraints on Taylor? It would have been better to have samples identical numbers for direct comparison - although I am happy that differences between the two population are so large that it was not critical.
- Productivity is key variable in the study. From reading, it appears that the key variable they use as a proxy for productivity is DNA concentration. A minor point, but this could be made plainer from the very beginning (e.g. line 147).
- The methods are somewhat unclear as to whether it is just the 'puck' of sediment that is melted, or the sediment+overlying ice (line 171) - in which case some of the species diversity may also come from the overlying frozen water in the hole?
Author Response
Response to Reviewer 1 Comments
This is an interesting study, using cryoconite holes on Dry Valley glaciers as a 'natural laboratory' to investigate the ecological concept of island species-area. It's a nice idea, and clearly written.
RESPONSE: Thank you for the constructive and thoughtful review.
Overall, I find the conclusions compelling, but have a couple of comments/suggestions below:
1.The main potential drawback to this study is the potential of sampling artefacts (the initial sampling of different masses of cryoconite) to confound the interpretation that species richness correlates with hole size. By the nature of sampling, more material was typically sampled, melted, and homogenised in larger relative to smaller holes prior to taking 0.3 g for DNA extraction and analysis. Could therefore the results be due to this artefact, rather than true hole size? The authors provide an answer to this point in the Appendix, and show that sediment mass does not have as good a linear model fit (R2 = 0.54) to species richness as hole size (R2 = 0.75). To further strengthen this, however, could they also conduct the same statistical analyses but only on those holes which had a diameter greater than that of the corer (as these would presumably have similar masses of sediment in the core but different diameters?) If these also show the same statistical trends (at least on Canada, where hopefully there would be enough
holes of sufficient size for this), then this would add another level of robustness to the key conclusions.
RESPONSE: The additional analysis suggested by the reviewer is a good idea. However, a linear model comparing g sediment collected with log10A of cryoconite holes on Canada Glacier for only the 21 samples from holes with diameters greater than 10 cm (the diameter of the corer) was still significant (t = 4.051, p < 0.001, R2adj = 0.44).
When the samples with the largest mass of sediments collected (> 80 g) were also excluded, to analyze only 13 samples from a smaller range of sediment mass collected (from holes with diameters ranging from 14-67 cm), the g sediment collected no longer significantly varied with log10A (t = 1.94, ns), nor with bacterial or eukaryotic diversity (bacteria: t = 2.04, ns; eukaryotes: t = 1.92, ns).
With the mass of sediment collected thus removed as a confounding covariate, log10A still significantly correlated with the diversity of both bacteria and eukaryotes (bacteria: t =4.52, p < 0.001, R2adj = 0.62; eukaryotes: t = 5.66, p < 0.001, R2adj = 0.72).
We have therefore updated Appendix A accordingly, inserting the following in lines 564-573:
“Additionally, we reanalyzed a subset of 13 of the samples from Canada Glacier across a range of hole sizes (diameters 14-67 cm) that represented a smaller range of sediment masses collected (15-80 g). Within this more comparable subset, g sediment collected did not significantly vary with log10A (t = 1.94, ns), nor with bacterial or eukaryotic diversity (bacteria: t = 2.04, ns; eukaryotes: t = 1.92, ns). With the mass of sediment collected thus removed as a confounding covariate, log10A still significantly correlated with the diversity of both bacteria and eukaryotes (bacteria: t = 4.52, p < 0.001, R2adj = 0.62; eukaryotes: t =
5.66, p < 0.001, R2adj = 0.72).”
2. I was also wondering why different numbers of holes were sampled on each of the two glaciers (23 on Canada, 11 on Taylor) - was this time constraints on Taylor? It would have been better to have samples identical numbers for direct comparison - although I am happy that differences between the two population are so large that it was not critical.
RESPONSE: The reviewer is correct that logistical support constraints limited the
number of samples we were able to collect from Taylor Glacier. We have added a sentence to clarify this point in lines 157-159 which reads, “Logistical constraints prevented the collection of as many samples from Taylor Glacier as from Canada Glacier.”
3. Productivity is key variable in the study. From reading, it appears that the key variable they use as a proxy for productivity is DNA concentration. A minor point, but this could be made plainer from the very beginning (e.g. line 147).
RESPONSE: We have updated Hypothesis 2 to clarify we hypothesize “2) that additional variation in the ISAR within each glacier would be related to measures of productivity such as DNA concentration because more productive holes would essentially function as biologically “larger” islands than their physical size would suggest [46];”
4. The methods are somewhat unclear as to whether it is just the 'puck' of sediment that is melted, or the sediment+overlying ice (line 171) - in which case some of the species diversity may also come from the overlying frozen water in the hole?
RESPONSE: We have clarified the wording in line 171 to now read, “In addition, the outer layer of ice and sediment was melted off each sampled sediment puck by using filtered, deionized water in the laboratory before each sample was melted and homogenized.”
Reviewer 2 Report
The work was carried out on the material of two glaciers in Antarctica. Despite the seemingly narrow specificity of this study, it raises many general biological and environmental issues that are important for the biosphere as a whole. The authors raised a number of specific questions and tried to find answers to them. Cryoconite holes were a model of isolated natural systems ("islands"). The work is well thought out methodically. The results are mathematically verified. The main conclusion is confirmation that the island species-area relationship (ISAR) is a positive association between the number of species and the area of ​​an island-like habitat. The assessment of microorganism diversity was carried out by DNA analysis. The genera of the main prokaryotes and eukaryotes have been identified.
There are many photosynthetic microorganisms among them, actinobacteria are also noted. Although the authors did not set themselves the task of isolating strains of microorganisms, it would be interesting to study the antibiotic activity of actinobacteria from these ecosystems of fierce interspecies competition.
The article can be published without changes.
Author Response
Response to Reviewer 2 Comments
The work was carried out on the material of two glaciers in Antarctica. Despite the seemingly narrow specificity of this study, it raises many general biological and environmental issues that are important for the biosphere as a whole. The authors raised a number of specific questions and tried to find answers to them. Cryoconite holes were a model of isolated natural systems ("islands"). The work is well thought out methodically. The results are mathematically verified. The main conclusion is confirmation that the island species-area relationship (ISAR) is a
positive association between the number of species and the area of an island-like habitat. The assessment of microorganism diversity was carried out by DNA analysis. The genera of the main prokaryotes and eukaryotes have been identified.
There are many photosynthetic microorganisms among them, actinobacteria are also noted. Although the authors did not set themselves the task of isolating strains of microorganisms, it would be interesting to study the antibiotic activity of actinobacteria from these ecosystems of fierce interspecies competition.
The article can be published without changes.
RESPONSE: Thank you for the constructive and thoughtful review.